# A Novel Anti-CD44 Variant 3 Monoclonal Antibody C_44_Mab-6 Was Established for Multiple Applications

**DOI:** 10.3390/ijms24098411

**Published:** 2023-05-07

**Authors:** Hiroyuki Suzuki, Kaishi Kitamura, Nohara Goto, Kenichiro Ishikawa, Tsunenori Ouchida, Tomohiro Tanaka, Mika K. Kaneko, Yukinari Kato

**Affiliations:** 1Department of Molecular Pharmacology, Tohoku University Graduate School of Medicine, 2-1 Seiryo-machi, Aoba-ku, Sendai 980-8575, Japan; kitamura.kaishi.s7@dc.tohoku.ac.jp (K.K.); s1930550@s.tsukuba.ac.jp (N.G.); ken.ishikawa.r3@dc.tohoku.ac.jp (K.I.); tsunenori.ouchida.d5@tohoku.ac.jp (T.O.); tomohiro.tanaka.b5@tohoku.ac.jp (T.T.); k.mika@med.tohoku.ac.jp (M.K.K.); 2Department of Antibody Drug Development, Tohoku University Graduate School of Medicine, 2-1 Seiryo-machi, Aoba-ku, Sendai 980-8575, Japan

**Keywords:** CD44, CD44 variant 3, monoclonal antibody, flow cytometry, immunohistochemistry

## Abstract

Cluster of differentiation 44 (CD44) promotes tumor progression through the recruitment of growth factors and the acquisition of stemness, invasiveness, and drug resistance. CD44 has multiple isoforms including CD44 standard (CD44s) and CD44 variants (CD44v), which have common and unique functions in tumor development. Therefore, elucidating the function of each CD44 isoform in a tumor is essential for the establishment of CD44-targeting tumor therapy. We have established various anti-CD44s and anti-CD44v monoclonal antibodies (mAbs) through the immunization of CD44v3–10-overexpressed cells. In this study, we established C_44_Mab-6 (IgG_1_, kappa), which recognized the CD44 variant 3-encoded region (CD44v3), as determined via an enzyme-linked immunosorbent assay. C_44_Mab-6 reacted with CD44v3–10-overexpressed Chinese hamster ovary (CHO)-K1 cells (CHO/CD44v3–10) or some cancer cell lines (COLO205 and HSC-3) via flow cytometry. The apparent *K*_D_ of C_44_Mab-6 for CHO/CD44v3–10, COLO205, and HSC-3 was 1.5 × 10^−9^ M, 6.3 × 10^−9^ M, and 1.9 × 10^−9^ M, respectively. C_44_Mab-6 could detect the CD44v3–10 in Western blotting and stained the formalin-fixed paraffin-embedded tumor sections in immunohistochemistry. These results indicate that C_44_Mab-6 is useful for detecting CD44v3 in various experiments and is expected for the application of tumor diagnosis and therapy.

## 1. Introduction

The cell surface glycoprotein known as cluster of differentiation 44 (CD44) is broadly expressed by epithelial, mesenchymal, and hematopoietic cells and is involved in adhesion to the extracellular matrix (ECM), lymphocyte homing, and lymphocyte activation [1]. A growing body of evidence reveals the critical roles of CD44 in tumor progression and metastasis [2,3]. The human CD44 gene consists of 19 exons, 10 of which are constant in all variants, and makes up the standard form of CD44 (CD44s) [4]. Furthermore, a large number of CD44 variants (CD44v) are generated due to alternative splicing. The CD44v consists of 10 constant exons in combination with the remaining 9 variant exons.

The translated CD44 usually receives post-translational modifications, such as *N*-/*O*-linked glycosylation or proteoglycans, including chondroitin sulfate, keratan sulfate, and heparan sulfate, which lead to further diversity in CD44 function [5,6,7,8]. Therefore, the molecular weights of CD44s and CD44v are 75–95 kDa and 100~250 kDa, respectively [5]. These CD44 isoforms have both overlapping and unique functions. Both CD44s and CD44v (pan-CD44) possess hyaluronic acid (HA)-binding motifs that promote interaction with the microenvironment, which mediates cellular homing, migration, adhesion, and proliferation [9].

CD44v is overexpressed in tumors, and it plays critical roles in the promotion of tumor invasion, metastasis, cancer-initiating properties [10], and resistance to therapies [2,11]. CD44v has the ability to bind to heparin-binding growth factors, including fibroblast growth factors (FGFs) [7]. These growth factors bind to a heparan sulfate side chain attached to the SGSG sequence encoded by variant exon 3 [7,12]. Heparan sulfate proteoglycans play critical roles in cell proliferation and motility through presenting the growth factors to receptors. Therefore, the CD44 variant exon 3-containing isoform (CD44v3) can recruit heparin-binding growth factors to their receptor and promote growth-factor-receptor-mediated signal transduction [13,14,15]. Furthermore, the v6-encoded region forms a complex with hepatocyte growth factor and its receptor MET, which is essential for activation [16]. The v8–10-encoded region interacts with a cystine–glutamate transporter (xCT) subunit and mediates the oxidative stress resistance through the regulation of the intracellular redox state [17].

Cancer stem cells (CSCs) exhibit important properties, such as self-renewal, resistance to therapy, and the promotion of tumor metastasis [18,19,20]. Several cell surface and intracellular proteins have been reported as CSC markers in hematopoietic malignancy and solid tumors [21,22]. Among them, CD44 has been identified as a CSC marker in several solid tumors. In breast cancer, the CD44^+^CD24^−/low^Lineage^−^ population was first shown to be 10- to 50-fold enriched with the ability to form tumors in immunodeficient mice relative to unfractionated tumor cells [23]. In head and neck squamous cell carcinoma (HNSCC), the CD44-high CSCs from HNSCC exhibited elevated migration, invasiveness, and stemness [24,25] and could form metastatic foci in the lungs of immunodeficient mice. In contrast, the CD44-low populations failed to form the metastatic proliferation [26]. In the above studies, anti-pan-CD44 monoclonal antibodies (mAbs) were used to isolate the CSCs from cultured cells and patient-derived tumor tissues. Furthermore, several CD44v-specific mAbs were reported to separate CSCs from colorectal cancer by using anti-CD44v6 [27] and anti-CD44v9 [28] mAbs. Therefore, specific mAbs against CD44s and CD44v are required for the isolation of CSCs and the analysis of their properties in detail.

By using the Cell-Based Immunization and Screening (CBIS) method, we established an anti-pan-CD44 mAb, namely, C_44_Mab-5 (IgG_1_, kappa) [29]. We further established another anti-pan-CD44 mAb, namely, C_44_Mab-46 (IgG_1_, kappa), via the immunization of CD44v3–10 ectodomain (CD44ec) [30]. Both C_44_Mab-5 and C_44_Mab-46 have been revealed to recognize the standard exon (1 to 5)-encoding sequences at the N-terminus [31,32,33]. Furthermore, both C_44_Mab-5 and C_44_Mab-46 are available for flow cytometry and immunohistochemical analyses in oral squamous cell carcinomas (OSCC) [29] and esophageal squamous cell carcinomas [30]. We further converted the mouse IgG_1_ subclass antibody (C_44_Mab-5) into an IgG_2a_ subclass antibody (5-mG_2a_) and further produced a defucosylated version (5-mG_2a_-f) by using fucosyltransferase 8-deficient ExpiCHO-S (BINDS-09) cells. The 5-mG_2a_-f exhibited *in vitro* antibody-dependent cellular cytotoxicity (ADCC) activity against OSCC cell lines (HSC-2 and SAS). Furthermore, the 5-mG_2a_-f suppressed the growth of the HSC-2 and SAS xenograft [34].

Recently, we established an anti-CD44v5 mAb [35] and an anti-CD44v6 mAb [36] via the CBIS method, an anti-CD44v7/8 mAb [37] via the immunization of CD44ec, and an anti-CD44v4 mAb via peptide immunization [38]. In this study, we developed a novel anti-CD44v3 mAb, namely, C_44_Mab-6 (IgG_1_, kappa), via the CBIS method and evaluated its applications, such as flow cytometry, Western blot, and immunohistochemical analyses.

## 2. Results

### 2.1. Development of C_44_Mab-6 as an anti-CD44v3 mAb

The CBIS method involves the immunization of antigen-overexpressed cells and high-throughput hybridoma screening by using flow cytometry. We prepared CD44v3–10-overexpressed Chinese hamster ovary (CHO)-K1 cells (CHO/CD44v3–10), as an immunogen (Figure 1). The cells were immunized into mice, and hybridomas were plated into 96-well plates. We next performed flow-cytometry-based, high-throughput screening to select the supernatants, which were positive for CHO/CD44v3–10 cells and negative for CHO-K1 cells. After the limiting dilution, anti-CD44 mAb-producing clones were finally established. Among them, C_44_Mab-6 (IgG_1_, kappa) was shown to recognize CD44 p231–250 peptide (AGWEPNEENEDERDRHLSFS), which corresponds to variant-3-encoded sequence (Figure 2 and Appendix A). In contrast, C_44_Mab-6 never recognized other extracellular region peptides of CD44v3-10. These results indicated that C_44_Mab-6 specifically recognizes the CD44 variant 3-encoded sequence.

### 2.2. The Reactivity of C_44_Mab-6 to CD44-Expressing Cells in Flow Cytometry

The reactivity of C_44_Mab-6 to CHO/CD44v3–10, CHO/CD44s, and CHO-K1 cells was investigated by using flow cytometry. C_44_Mab-6 dose-dependently recognized CHO/CD44v3–10 cells (Figure 3A). In contrast, C_44_Mab-6 recognized neither CHO/CD44s (Figure 3B) nor CHO-K1 (Figure 3C) cells. C_44_Mab-46, which is an anti-pan-CD44 mAb [30], recognized both CHO/CD44v3–10 and CHO/CD44s cells (Appendix A). We next examined the reactivity of C_44_Mab-6 to a colorectal cancer cell line (COLO205) and an OSCC cell line (HSC-3). COLO205 was selected in this study from various cancer cell lines because C_44_Mab-6 showed very high reactivity to it. Furthermore, HSC-3 was selected because HNSCC was shown to be the second highest CD44-expressing cancer type in the Pan-Cancer Atlas [39]. C_44_Mab-6 could recognize a colorectal cancer cell line COLO205 (Figure 3D) and an oral squamous cell line HSC-3 (Figure 3E) in a dose-dependent manner.

### 2.3. The Binding Affinity of C_44_Mab-6 to CD44-Expressing Cells

The binding affinity of C_44_Mab-6 to CHO/CD44v3–10, COLO205, and HSC-3 was determined by using flow cytometry. As shown in Figure 4, the *K*_D_ of CHO/CD44v3–10 COLO205 and HSC-3 was determined as 1.5 × 10^−9^ M, 6.3 × 10^−9^ M, and 1.9 × 10^−9^ M, respectively. These results indicated that C_44_Mab-6 possesses a high affinity for CD44v3–10 and endogenous CD44v3-expressing cells.

### 2.4. Western Blot Analysis

Western blot analysis was preformed to investigate the specificity of C_44_Mab-6. As shown in Figure 5A, C_44_Mab-6 detected CD44v3–10 as showing more than 180-kDa bands. However, C_44_Mab-6 did not detect any bands from lysates of CHO-K1 and CHO/CD44s cells at more than 48-kDa. An anti-pan-CD44 mAb, namely, C_44_Mab-46, recognized the lysates from both CHO/CD44s (~75 kDa) and CHO/CD44v3–10 (>180 kDa) (Figure 5B). These results indicated that C_44_Mab-6 specifically detects exogenous CD44v3–10 but not CD44s.

### 2.5. Immunohistochemical Analysis by Using C_44_Mab-6 against Tumor Tissues

Immunohistochemical analysis against the formalin-fixed paraffin-embedded (FFPE) sections of OSCC was conducted to assess the availability of C_44_Mab-6. We used sequential sections of OSCC tissue microarray and compared the staining patterns of C_44_Mab-6 and C_44_Mab-46. Clear membranous staining was observed for C_44_Mab-6 and C_44_Mab-46 in a well-differentiated OSCC section (Figure 6A,B). Figure 6C,D showed an OSCC section with the stromal invaded phenotype. C_44_Mab-6 strongly stained stromal-invaded OSCC and could clearly distinguish tumor cells from the surrounding stroma cells (Figure 6C). In contrast, C_44_Mab-46 stained both invaded tumor and stromal cells (Figure 6D). In Figure 6E and F, C_44_Mab-6 partially stained tumor cells but not stromal cells (Figure 6E). In contrast, C_44_Mab-46 mainly stained stromal cells (Figure 6F). We summarized the data of immunohistochemical analysis of CD44 expression in tumor cells in Table 1; C_44_Mab-6 stained 44 out of 50 (88%) cases of OSCC. We also stained FFPE sections of colorectal cancer tissue microarray and found that C_44_Mab-6 stained 7 out of 40 (18%) cases (Appendix A). However, the C_44_Mab-6 reactivity was faint and partly localized compared to that of C_44_Mab-46 (Appendix A). These results indicated that C_44_Mab-6 is useful for the immunohistochemical analysis of FFPE tumor sections.

## 3. Discussion

In this study, we developed C_44_Mab-6 by using the CBIS method (Figure 1) and determined its epitope as a variant 3-encoded region of CD44 (Figure 2 and supplemental Appendix A). Then, we showed the usefulness of C_44_Mab-6 for multiple applications, including flow cytometry (Figure 3 and Figure 4), Western blotting (Figure 5), and immunohistochemistry (Figure 6).

An anti-CD44v3 mAb (clone 3G5) was previously developed and widely used for various applications [40]. The 3G5 was developed by the immunization of COS1-produced CD44v3-10-Fc protein. The specificity to the exon was determined via indirect immunofluorescent staining of COS1 cells which expressed CD44v3–10, CD44v6–10, CD44v7–10, CD44v8–10, and CD44v10 [40]. Therefore, the 3G5 is thought to recognize the peptide or glycopeptide structure of CD44v3. However, the detailed binding epitope of 3G5 has not been determined. As shown in Appendix A, C_44_Mab-6 recognized CD44 p231–250 peptide (AGWEPNEENEDERDRHLSFS) but not CD44 p241–260 peptides (DERDRHLSFSGSGIDDDEDF). The underlined SGSG sequence is a heparan sulfate-modified sequence in the variant-3-encoded region [7,12]. Therefore, the recognition of C_44_Mab-6 is probably not affected by the heparan sulfate modification.

Head and neck cancers are derived mainly from the oral cavity, larynx, pharynx, and nasal cavity [41]. SCC is the common type. As shown in Figure 6, C_44_Mab-6 clearly stained the membrane of OSCC and recognized a human OSCC cell line, namely, HSC-3 (Figure 3 and Figure 4). The CD44v3-high and aldehyde dehydrogenase-1 (ALDH1)-high population of HSC-3 exhibited a potent tumorigenic potential in immunodeficient NOD/SCID mice [42]. The population showed increased stemness-related transcriptional factors, including OCT4, SOX2, and NANOG [42]. In a future study, we will investigate the application of C_44_Mab-6 to isolate cancer stem-like cells from cancer cell lines and/or OSCC tissues. We will further establish the strategy to deplete the cancer stem-like cells for tumor therapy. We have just started the cDNA cloning of C_44_Mab-6 heavy and light chains for therapeutic application. We have investigated the antitumor activity by using class-switched and defucosylated IgG_2a_ mAbs [34,43,44,45,46,47,48,49]. The defucosylated IgG_2a_ mAbs can be produced by fucosyltransferases 8-deficient CHO-K1 cells, exhibited potent ADCC activity in vitro, and suppressed the xenograft growth [34,43,44,45,46,47,48,49]. Therefore, the production of defucosylated C_44_Mab-6 is one of the strategies to evaluate antitumor activity in vivo.

The HNSCC treatments include surgery, chemotherapy, radiotherapy, molecular targeted therapy, immunotherapy, or a combination of these modalities [50]. Despite the progress of the therapies, drug resistance and metastasis are still the main causes of death [51]. In a preclinical study, a pan-CD44 mAb was applied to the novel modalities, including near-infrared photoimmunotherapy (NIR-PIT). The CD44 mAb–photoactivatable dye IRDye700DX conjugate exhibited significant antitumor effects after the NIR-light exposure against CD44-expressing OSCC [52]. However, a pan-CD44 mAb, namely, C_44_Mab-46, recognized not only tumor cells but also stromal tissues (Figure 6D,F) and probably immune cells, which are important for antitumor immunity. Therefore, CD44v is a more rational tumor antigen for NIR-PIT, which could be a new modality for OSCC with locoregional recurrence.

Zen et al. established a unique mAb (clone C3H7), which recognized the basolateral membranes of epithelium and inhibited both the adhesion of epithelial cells to immobilized CD11b/CD18 and the transepithelial migration of leukocytes [53]. CD11b/CD18, also known as Macrophage-1 antigen, is a leukocyte integrin that is essential for firm adhesion to epithelial cells and the transepithelial migration of leukocytes [54]. However, the receptor of CD11b/CD18 on epithelial cells has not been identified. They revealed that the antigen of C3H7 is CD44v3, which specifically binds to CD11b/CD18 through its heparan sulfate moieties [53]. The C3H7 antigen was increased via treatment with pro-inflammatory cytokine, including interferon-γ and tumor necrosis factor-α in epithelial monolayers [53], which supports the previous finding that CD44v3 is increased in inflammatory diseases, including ulcerative colitis [55]. C_44_Mab-6 also recognized the basolateral surface of colorectal cancer cells (Appendix A). Further investigations are required for the relationship between CD44v3 expression and the transepithelial migration of leukocytes. The study could provide the basis for the development of novel therapeutic applications of anti-CD44v3 mAbs, including C_44_Mab-6.

Chimeric antigen receptor T-cell (CAR-T) therapies have been developed for a variety of hematopoietic malignances and solid tumors [56]. CAR-T cells have demonstrated remarkable success in treating CD19^+^ B cell leukemias [57]. However, CAR-T therapy for acute myeloid leukemia (AML) has been elusive because of target restriction and phenotypic heterogeneity [58]. Mutations of the FMS-like tyrosine kinase 3 (*FLT3*) and DNA methyltransferase 3A (*DNMT3A*) genes were identified as common driver mutations associated with poor prognosis of AML patients [59]. Tang et al. showed that AML cells expressed high levels of CD44 mRNA, and the expression of AML-derived FLT3 and DNMT3A mutants promote the transcription of CD44 mRNA through suppression of CpG island methylation in the CD44 promoter [60]. They also found that AML patients with *FLT3* or *DNMT3A* mutations had higher expression of CD44v6 compared to normal specimens. Furthermore, they showed that CD44v6 CAR-T cells exhibited potent anti-leukemic effects [60]. Therefore, CD44v6 is thought to be a rational antigen of CAR-T therapy for AML with *FLT3* or *DNMT3A* mutations.

Since CD44 mRNA is upregulated in AML, there is a possibility that other CD44 variants are also transcribed and expressed in AML. In a humanized mouse model of chronic myeloid leukemia (CML) progression from chronic phase to blast crisis, a CD44 variant (CD44v8–10) was elevated, which is required for the maintenance of stemness [61]. Although we have examined the reactivity of C_44_Mab-6 against cell lines derived from hematopoietic malignancy and found increased reactivity in several cell lines, further studies are required for the selective expression of CD44v3 in leukemia cells, but not in hematopoietic stem cells, to ensure its safety as a CAR-T antigen.

## 4. Materials and Methods

### 4.1. Cell Lines

COLO205 (a human colorectal cancer cell line) was obtained from the Cell Resource Center for Biomedical Research Institute of Development, Aging and Cancer at Tohoku University (Sendai, Japan). HSC-3 (a human OSCC cell line) and LN229 (a human glioblastoma cell line) were obtained from the Japanese Collection of Research Bioresources (Osaka, Japan). P3X63Ag8U.1 (P3U1: a mouse multiple myeloma) and CHO-K1 cell lines were obtained from the American Type Culture Collection (ATCC, Manassas, VA, USA). HSC-3 and LN229 were cultured in Dulbecco’s Modified Eagle Medium (DMEM) (Nacalai Tesque, Inc., Kyoto, Japan), supplemented with 10% (*v*/*v*) heat-inactivated fetal bovine serum (FBS; Thermo Fisher Scientific, Inc., Waltham, MA, USA), 100 U/mL of penicillin, 100 μg/mL streptomycin, and 0.25 μg/mL amphotericin B. CHO-K1, COLO205, and P3U1 were cultured in Roswell Park Memorial Institute (RPMI)-1640 medium (Nacalai Tesque, Inc.) supplemented with 10% FBS and antibiotics, as indicated above. All the cells were grown in a humidified incubator at 37 °C with 5% CO_2_.

### 4.2. Construction of Expression Plasmids and Stable Transfectants

By using LN229 cDNA as a template, CD44s cDNA was amplified by using HotStar HiFidelity Polymerase Kit (Qiagen Inc., Hilden, Germany). We obtained CD44v3–10 cDNA from the RIKEN BRC through the National Bio-Resource Project of the MEXT, Japan. The CD44s and CD44v3–10 cDNAs were cloned into pCAG-Ble-ssPA16 vector with signal sequence and N-terminal PA16 tag (GLEGGVAMPGAEDDVV) [29,62,63,64,65], which is detected by NZ-1, which was originally developed as an anti-human podoplanin mAb [66,67,68,69,70,71,72,73,74,75,76,77,78,79,80,81]. The pCAG-Ble/PA16-CD44s and pCAG-Ble/PA16-CD44v3–10 vectors were transfected into CHO-K1 cells by using a Neon transfection system (Thermo Fisher Scientific, Inc.), and CHO/CD44s and CHO/CD44v3–10 were finally established, as described previously [36].

### 4.3. Production of Hybridomas

All animal experiments were approved by the Animal Care and Use Committee of Tohoku University (Permit number: 2019NiA-001). The female BALB/c mice (CLEA Japan, Tokyo, Japan) were intraperitoneally immunized with CHO/CD44v3–10 (1 × 10^8^ cells) with Imject Alum (Thermo Fisher Scientific Inc.) as an adjuvant. The three additional immunizations per week and a booster injection were performed two days before the harvest of the spleen cells. The hybridomas were produced via the fusion of splenocytes and P3U1 cells by using polyethylene glycol 1500 (PEG1500; Roche Diagnostics, Indianapolis, IN, USA). The supernatants, which were positive for CHO/CD44v3–10 cells and negative for CHO-K1 cells, were selected by using the SA3800 Cell Analyzers (Sony Corp., Tokyo, Japan).

### 4.4. ELISA

We obtained fifty-eight peptides, which cover the extracellular domain of CD44v3–10 [31], from Sigma-Aldrich Corp. (St. Louis, MO, USA). We immobilized them on Nunc Maxisorp 96-well immunoplates (Thermo Fisher Scientific, Inc.) at 1 µg/mL for 30 min at 37 °C. Immunoplate washing was performed by using HydroSpeed Microplate Washer (Tecan, Zürich, Switzerland) with phosphate-buffered saline (PBS) containing 0.05% (*v*/*v*) Tween 20 (PBST; Nacalai Tesque, Inc.). After the blocking with 1% (*w*/*v*) bovine serum albumin (BSA) in PBST for 30 min at 37 °C, C_44_Mab-6 (10 µg/mL) was added to each well. Then, the wells were further incubated with anti-mouse immunoglobulins–peroxidase conjugate (1:2000 diluted; Agilent Technologies Inc., Santa Clara, CA, USA) for 30 min at 37 °C. One-Step Ultra TMB (Thermo Fisher Scientific Inc.) was used for enzymatic reactions. An iMark microplate reader (Bio-Rad Laboratories, Inc., Berkeley, CA, USA) was used to measure the optical density at 655 nm.

### 4.5. Flow Cytometry

CHO-K1, CHO/CD44s, CHO/CD44v3–10, COLO205, and HSC-3 cells were obtained by using 0.25% trypsin and 1 mM ethylenediamine tetraacetic acid (EDTA; Nacalai Tesque, Inc.). The cells were treated with C_44_Mab-6, C_44_Mab-46, or blocking buffer (control) (0.1% BSA in PBS) for 30 min at 4 °C. Then, the cells (1 × 10^5^ cells/sample) were treated with anti-mouse IgG conjugated with Alexa Fluor 488 (1:2000; Cell Signaling Technology, Inc., Danvers, MA, USA) for 30 min at 4 °C. The data were analyzed by using the SA3800 Cell Analyzer and SA3800 software ver. 2.05 (Sony Corp.).

### 4.6. Determination of Dissociation Constant (K_D_) via Flow Cytometry

In CHO/CD44v3–10 cells, we prepared from 130 to 0.008 nM (diluted by 1/2) of C_44_Mab-6. In COLO201 and HSC-3 cells, we prepared from 1300 to 0.08 nM (diluted by 1/2) of C_44_Mab-6. The serially diluted C_44_Mab-6 was suspended with 2 × 10^5^ cells. Then, the cells were incubated with anti-mouse IgG conjugated with Alexa Fluor 488 (1:200). BD FACSLyric and BD FACSuite software version 1.3 (BD Biosciences, Franklin Lakes, NJ, USA) were used for the fluorescence data analyses. The *K*_D_ was determined by the fitting binding isotherms to built-in one-site binding models of GraphPad Prism 8 (GraphPad Software, Inc., La Jolla, CA, USA).

### 4.7. Western Blot Analysis

The 10 μg of cell lysates were subjected to SDS-polyacrylamide gel for electrophoresis by using polyacrylamide gels (5–20%; FUJIFILM Wako Pure Chemical Corporation, Osaka, Japan). The separated proteins were transferred onto polyvinylidene difluoride (PVDF) membranes (Merck KGaA, Darmstadt, Germany). The blocking was performed by using 4% skim milk (Nacalai Tesque, Inc.) in PBST. The membranes were incubated with 10 μg/mL of C_44_Mab-6, 10 μg/mL of C_44_Mab-46, or 1 μg/mL of an anti-β-actin mAb (clone AC-15; Sigma-Aldrich Corp.) and then incubated with peroxidase-conjugated anti-mouse immunoglobulins (diluted 1:1000; Agilent Technologies, Inc.). Finally, the signals were enhanced by using a chemiluminescence reagent, ImmunoStar LD (FUJIFILM Wako Pure Chemical Corporation), and were detected by a Sayaca-Imager (DRC Co., Ltd., Tokyo, Japan).

### 4.8. Immunohistochemical Analysis

FFPE sections of OSCC tissue array (OR601c) and colorectal carcinoma tissue array (CO483a) were purchased from US Biomax Inc. (Rockville, MD, USA). The tissue arrays were autoclaved in EnVision FLEX Target Retrieval Solution High pH (Agilent Technologies, Inc.) for 20 min. After blocking with SuperBlock T20 (Thermo Fisher Scientific, Inc.), the sections were incubated with C_44_Mab-6 (1 μg/mL) and C_44_Mab-46 (1 μg/mL) for 1 h at room temperature. The sections were further incubated with the EnVision+ Kit for mouse (Agilent Technologies Inc.) for 30 min. Then, a chromogenic reaction using 3,3′-diaminobenzidine tetrahydrochloride (DAB; Agilent Technologies Inc.) was conducted. Hematoxylin (FUJIFILM Wako Pure Chemical Corporation) was used for the counterstaining. To examine the sections and obtain images, we used Leica DMD108 (Leica Microsystems GmbH, Wetzlar, Germany).

## Figures and Tables

**Figure 1 ijms-24-08411-f001:**
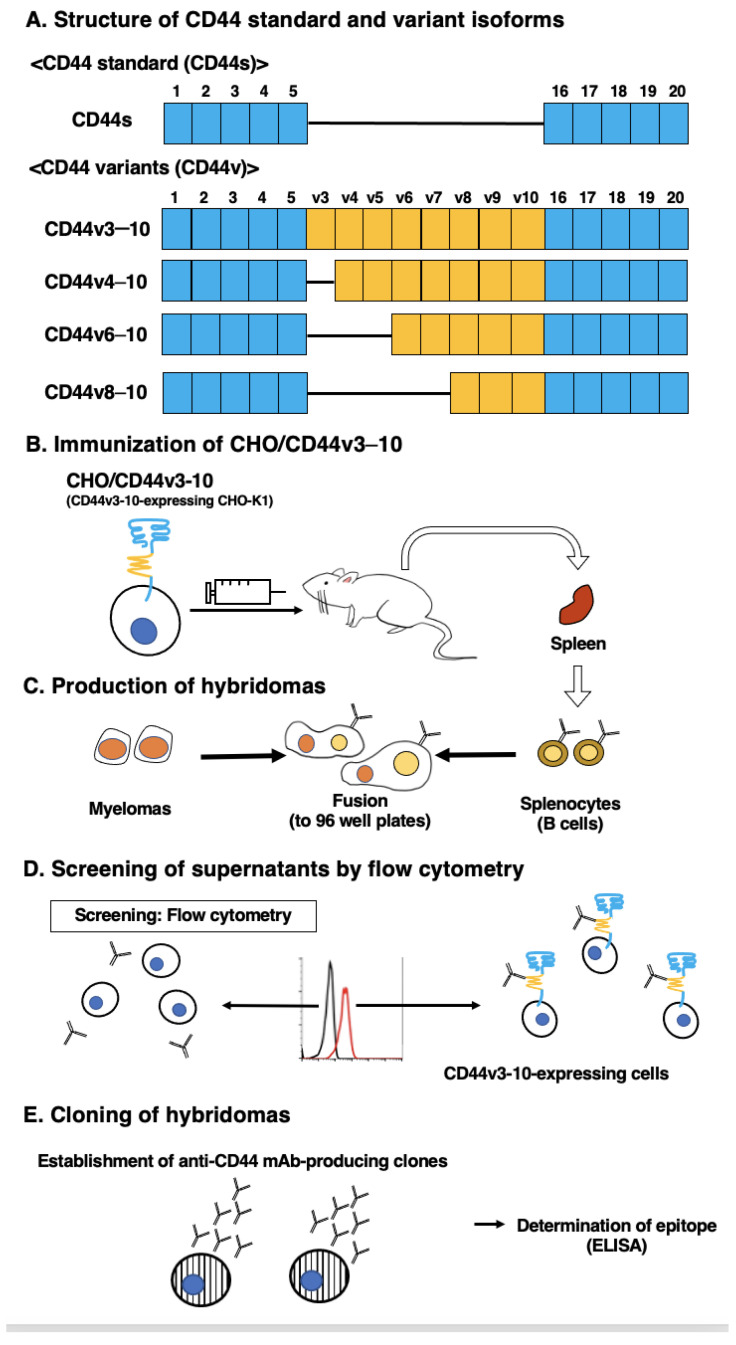
Anti-human CD44 mAbs production. (**A**) The CD44 structure. CD44s mRNA is assembled by the first five (1 to 5) and the last five (16 to 20) exons and translates CD44s. CD44v mRNAs are generated by the alternative splicing of variant exons and translate multiple CD44v isoforms, such as CD44v3-10, CD44v4-10, CD44v6-10, and CD44v8-10. (**B**) BALB/c mice were intraperitoneally immunized with CHO/CD44v3–10 cells. (**C**) The hybridomas were produced via fusion of the splenocytes and P3U1 cells. (**D**) The flow cytometry-mediated screening was conducted by using parental CHO-K1 and CHO/CD44v3–10 cells. (**E**) After cloning and additional screening, a clone C_44_Mab-6 (IgG_1_, kappa) was established. Finally, the binding epitope was determined via enzyme-linked immunosorbent assay (ELISA) by using peptides, which cover the extracellular domain of CD44v3–10.

**Figure 2 ijms-24-08411-f002:**
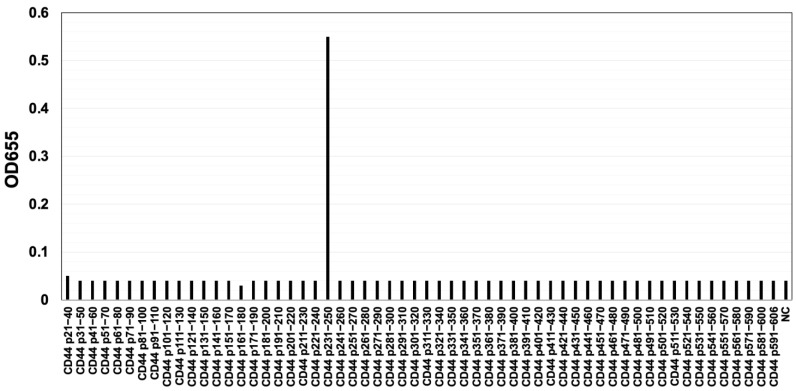
Determination of C_44_Mab-6 epitope by ELISA. The synthesized peptides, which cover the CD44v3–10 extracellular domain, were immobilized on immunoplates. The plates were incubated with C_44_Mab-6, followed by incubation with peroxidase-conjugated anti-mouse immunoglobulins. Optical density was measured at 655 nm. The CD44 p231–250 sequence (AGWEPNEENEDERDRHLSFS) corresponds to the variant 3-encoded sequence. ELISA: enzyme-linked immunosorbent assay. NC: negative control (solvent; DMSO in PBS).

**Figure 3 ijms-24-08411-f003:**
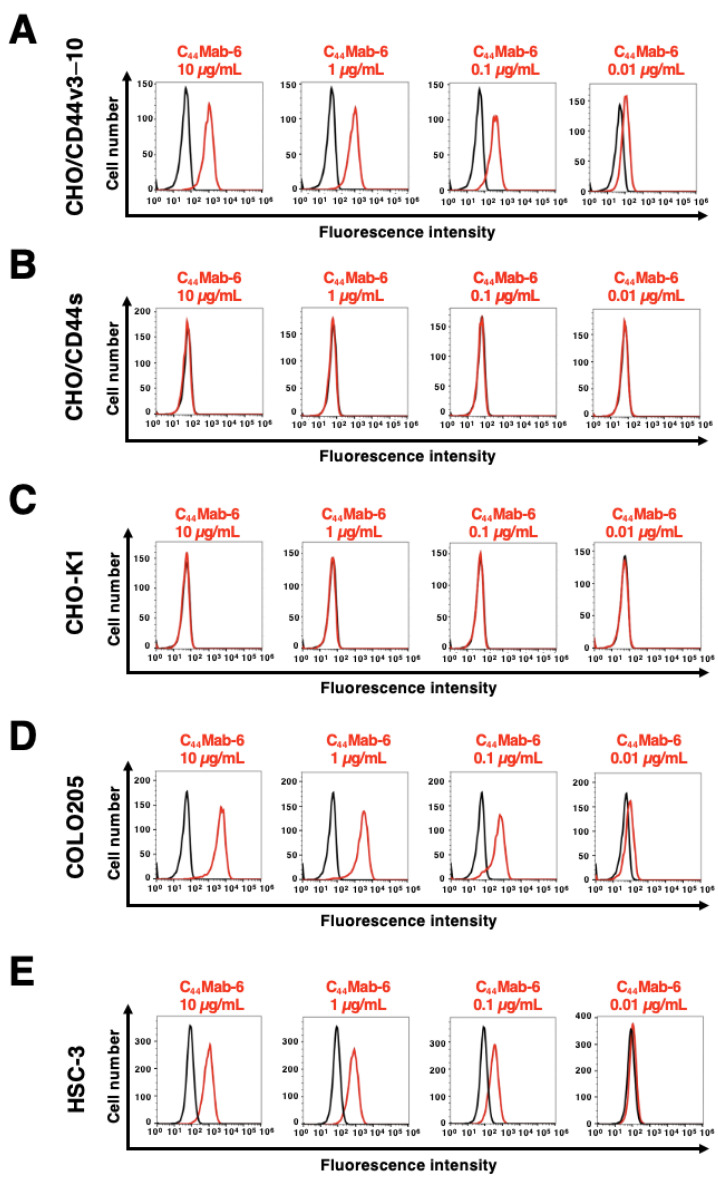
The reactivity of C_44_Mab-6 to CD44-expressing cells in flow cytometry. CHO/CD44v3–10 (**A**), CHO/CD44s (**B**), CHO-K1 (**C**), COLO205 (**D**), and HSC-3 (**E**) cells were treated with C_44_Mab-6 at 0.01–10 µg/mL, followed by treatment with anti-mouse IgG conjugated with Alexa Fluor 488 (Red line). Black line: negative control (blocking buffer).

**Figure 4 ijms-24-08411-f004:**
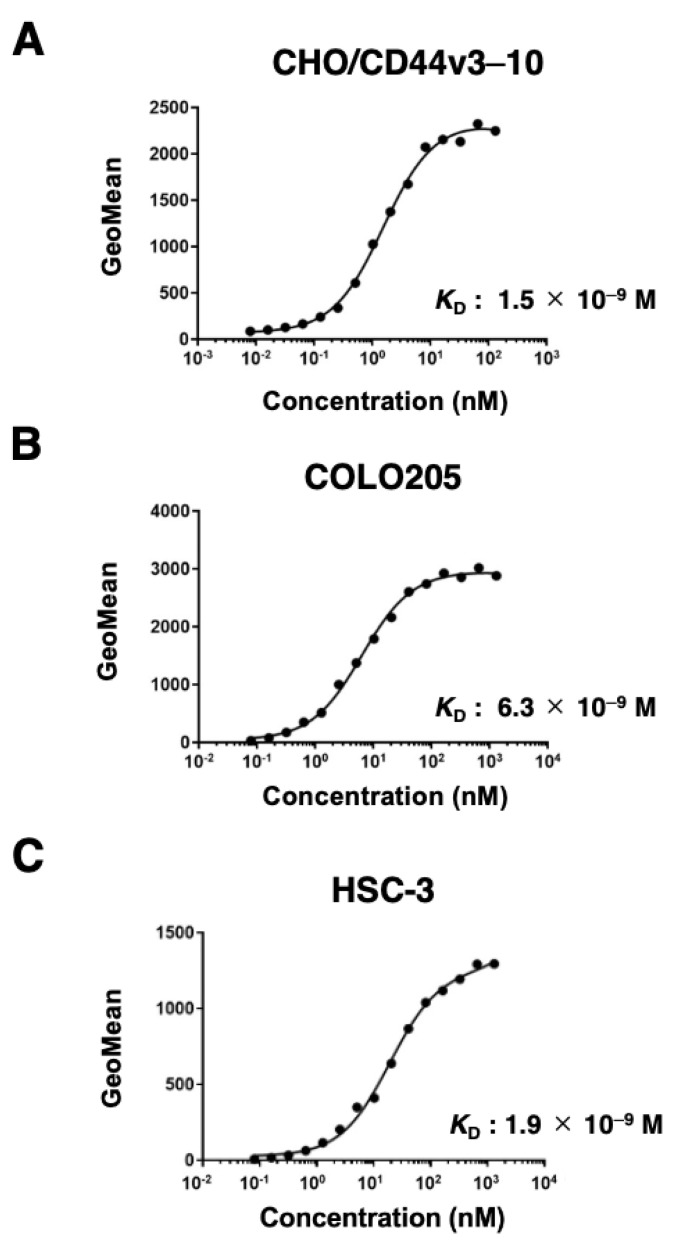
The binding affinity of C_44_Mab-6 to CD44-expressing cells. CHO/CD44v3–10 (**A**), COLO205 (**B**), and HSC-3 (**C**) cells were suspended in C_44_Mab-6 at the indicated concentrations. Cells were incubated with Alexa Fluor 488-conjugated secondary antibody. Fluorescence data were collected, and the apparent dissociation constant (*K*_D_) was calculated by using GraphPad Prism 8.

**Figure 5 ijms-24-08411-f005:**
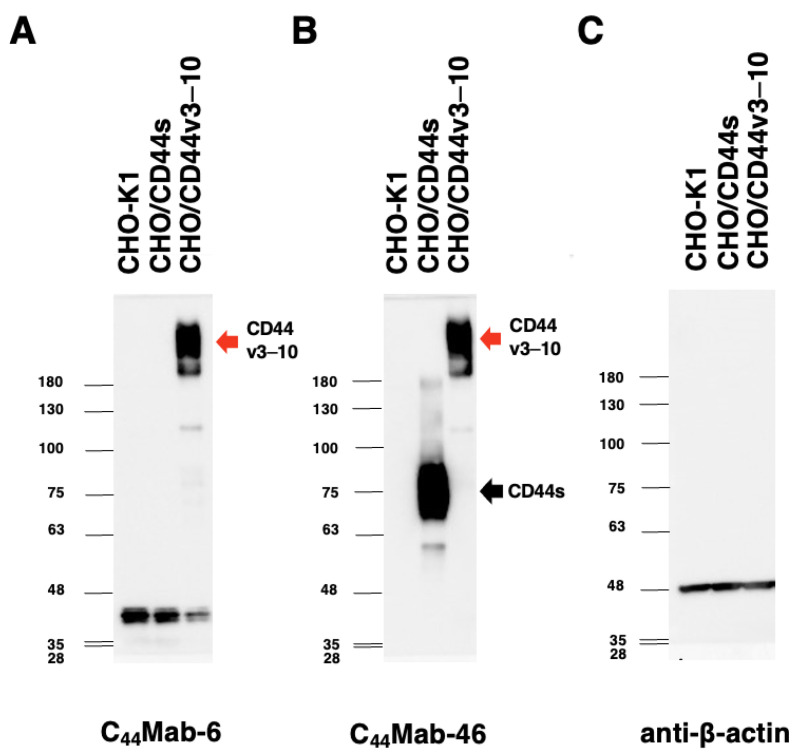
Western blot analysis by using C_44_Mab-6. The cell lysates of CHO-K1, CHO/CD44s, and CHO/CD44v3–10 (10 µg) were electrophoresed and transferred onto polyvinylidene fluoride membranes. The membranes were incubated with 10 µg/mL of C_44_Mab-6 (**A**), 10 µg/mL of C_44_Mab-46 (**B**), and 1 µg/mL of an anti-β-actin mAb (**C**). Then, the membranes were incubated with anti-mouse immunoglobulins conjugated with peroxidase. The black arrow indicates the CD44s (~75 kDa). The red arrows indicate the CD44v3–10 (>180 kDa).

**Figure 6 ijms-24-08411-f006:**
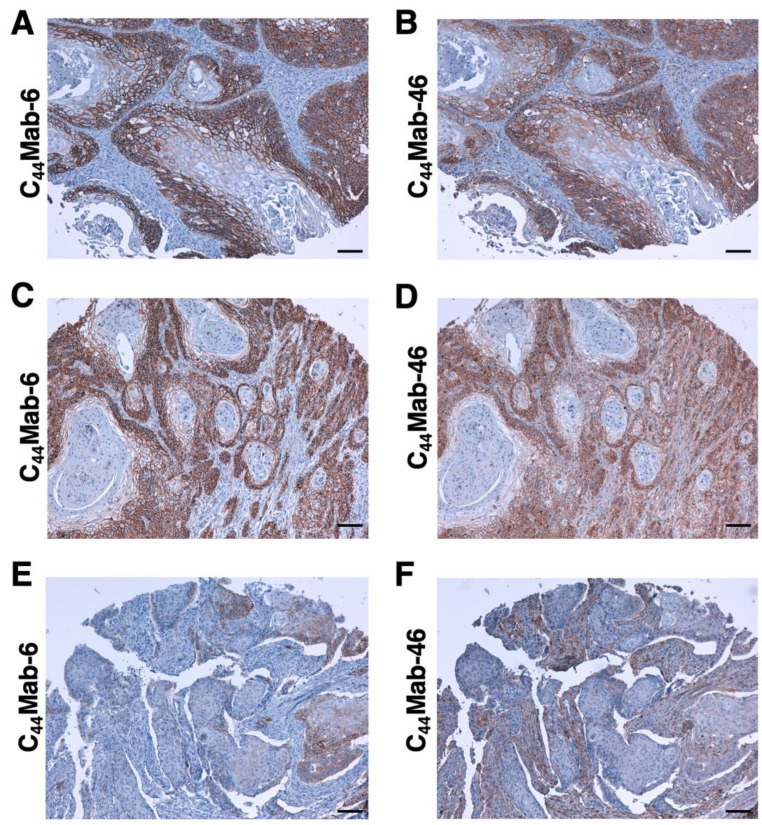
Immunohistochemical analysis by using C_44_Mab-6 and C_44_Mab-46 against OSCC tissues. After antigen retrieval, serial sections of OSCC tissue array (Catalog number: OR601c) were incubated with 1 μg/mL of C44Mab-6 (**A**,**C**,**E**) or 1 μg/mL of C44Mab-46 (**B**,**D**,**F**), followed by treatment with the Envision+ kit. The chromogenic reaction was conducted by using 3,3′-diaminobenzidine tetrahydrochloride (DAB). The counterstaining was performed by using hematoxylin. Scale bar = 100 μm.

**Table 1 ijms-24-08411-t001:** Immunohistochemical analysis by using C_44_Mab-6 and C_44_Mab-46 against OSCC tissue arrays.

No.	Age	Sex	Anatomic Site	Pathology Diagnosis	TNM	Grade	Type	C_44_Mab-6	C_44_Mab-46
1	78	M	Tongue	SCC of tongue	T2N0M0	1	Malignant	++	+
2	40	M	Tongue	SCC of tongue	T2N0M0	1	Malignant	++	++
3	75	F	Tongue	SCC of tongue	T2N0M0	1	Malignant	+	+
4	35	F	Tongue	SCC of tongue	T2N0M0	1	Malignant	++	++
5	61	M	Tongue	SCC of tongue	T2N0M0	1	Malignant	+++	+++
6	41	F	Tongue	SCC of tongue	T2N0M0	1	Malignant	+	+
7	64	M	Tongue	SCC of right tongue	T2N2M0	1	Malignant	++	++
8	76	M	Tongue	SCC of tongue	T1N0M0	1	Malignant	++	++
9	50	F	Tongue	SCC of tongue	T2N0M0	1	Malignant	++	++
10	44	M	Tongue	SCC of tongue	T2N1M0	1	Malignant	+++	+++
11	53	F	Tongue	SCC of tongue	T1N0M0	1	Malignant	++	++
12	46	F	Tongue	SCC of tongue	T2N0M0	1	Malignant	+	+
13	50	M	Tongue	SCC of root of tongue	T3N1M0	1	Malignant	+++	+
14	36	F	Tongue	SCC of tongue	T1N0M0	1	Malignant	+++	+++
15	63	F	Tongue	SCC of tongue	T1N0M0	1	Malignant	++	+
16	46	M	Tongue	SCC of tongue	T2N0M0	1	Malignant	+++	-
17	58	M	Tongue	SCC of tongue	T2N0M0	1	Malignant	+	+
18	64	M	Lip	SCC of lower lip	T1N0M0	1	Malignant	+++	+++
19	57	M	Lip	SCC of lower lip	T2N0M0	1	Malignant	+++	+++
20	61	M	Lip	SCC of lower lip	T1N0M0	1	Malignant	++	++
21	60	M	Gum	SCC of gum	T3N0M0	1	Malignant	+	+
22	60	M	Gum	SCC of gum	T1N0M0	1	Malignant	+++	+++
23	69	M	Gum	SCC of upper gum	T3N0M0	1	Malignant	++	++
24	53	M	Bucca cavioris	SCC of bucca cavioris	T2N0M0	1	Malignant	+	+
25	55	M	Bucca cavioris	SCC of bucca cavioris	T1N0M0	1	Malignant	++	+
26	58	M	Tongue	SCC of base of tongue	T1N0M0	1	Malignant	++	++
27	63	M	Oral cavity	SCC	T1N0M0	1	Malignant	++	++
28	48	F	Tongue	SCC of tongue	T1N0M0	1–2	Malignant	++	+
29	80	M	Lip	SCC of lower lip	T1N0M0	1–2	Malignant	+++	+++
30	77	M	Tongue	SCC of base of tongue	T2N0M0	1–2	Malignant	+++	++
31	59	M	Tongue	SCC of tongue	T2N0M0	2	Malignant	+	-
32	77	F	Tongue	SCC of tongue	T1N0M0	2	Malignant	++	++
33	56	M	Tongue	SCC of root of tongue	T2N1M0	2	Malignant	+	+
34	60	M	Tongue	SCC of tongue	T2N1M0	2	Malignant	++	++
35	62	M	Tongue	SCC of tongue	T2N0M0	2	Malignant	+++	++
36	67	F	Tongue	SCC of tongue	T2N0M0	2	Malignant	+++	++
37	47	F	Tongue	SCC of tongue	T2N0M0	2	Malignant	+++	+++
38	37	M	Tongue	SCC of tongue	T2N1M0	2	Malignant	-	-
39	55	F	Tongue	SCC of tongue	T2N0M0	2	Malignant	++	+
40	56	F	Bucca cavioris	SCC of bucca cavioris	T2N0M0	2	Malignant	+++	+
41	49	M	Bucca cavioris	SCC of bucca cavioris	T1N0M0	2	Malignant	-	-
42	45	M	Bucca cavioris	SCC of bucca cavioris	T2N0M0	2	Malignant	-	-
43	42	M	Bucca cavioris	SCC of bucca cavioris	T3N0M0	2	Malignant	+++	++
44	44	M	Jaw	SCC of right drop jaw	T1N0M0	2	Malignant	++	+++
45	40	F	Tongue	SCC of base of tongue	T2N0M0	2	Malignant	-	++
46	49	M	Bucca cavioris	SCC of bucca cavioris	T1N0M0	2	Malignant	+++	+++
47	56	F	Tongue	SCC of base of tongue	T2N0M0	3	Malignant	-	+
48	42	M	Bucca cavioris	SCC of bucca cavioris	T3N0M0	3	Malignant	+++	+++
49	87	F	Face	SCC of left face	T2N0M0	3	Malignant	+	+
50	50	M	Gum	SCC of gum	T2N0M0	3	Malignant	-	-

-: No stain; +: Weak intensity; ++: Moderate intensity; +++: Strong intensity.

## Data Availability

The data presented in this study are available in the article and supplementary material.

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
