# Peer review of "A Novel Anti-CD44 Variant 3 Monoclonal Antibody C44Mab-6 Was Established for Multiple Applications"

_ijms, 2023, doi:10.3390/ijms24098411_

Round 1
Reviewer 1 Report
In this manuscript, the authors developed C44Mab-6 from CD44v3-10 immunization and identified its epitope at CD44p231-250 by ELISA. Subsequently, the Mab was characterized using flow cytometry, western blot, and immunohistochemistry for its specificity and affinity.
The manuscript is well written with clear experimental design. However, there are six papers already published that are almost identical, focusing on the examination of individual Mab clones derived from CD44v3-10 immunization. While CD44Mab-6 appears to target a different epitope (CD44p231-250), it remains uncertain of the novelty of this manuscript. Having the studies mentioned in the discussion section (214-221, 222-232, 233-247, 248-261) or including a panel of clones would strengthen the quality of the manuscript.
Reviewer 2 Report
The manuscript demonstrates the development of a novel monoclonal antibody C44Mab-6, which targets CD44v3-10, using well-established cell-based immunization and screening method. Potential applications of C44Mab-6 were then evaluated by flow cytometry, western blot and immunohistochemical analysis. The presented data show promising future uses of C44Mab-6, which may facilitate the understanding of the progression of CD44v3-related diseases. Comments and suggestions of the reviewer are as followed:
1. The second sentence of first paragraph in the introduction section needs to be rephrased (A growing body …………metastasis).
2. What is the rationale of using COLO205 and HSC-3? Any hypothesis on why binding affinity on HSC-3 is better than that on COLO205?
3. Subtitle order under section 2 is not consistent, e.g. from 2.3 to 3.4.
4. In section 3.5 (should be section 2.5?), why different concentrations of C44Mab-6 and C44Mab-46 were applied?
Overall, the language is ok, but a few errors have been noticed, which I have pinpointed in the comments.
Round 2
Reviewer 1 Report
N/A
Author Response
Thank you very much.